# Out of the Box: Reasoning with Graph Convolution Nets for Factual Visual Question Answering

**Medhini Narasimhan, Svetlana Lazebnik, Alexander G. Schwing**
University of Illinois Urbana-Champaign
{medhini2, slazebni, aschwing}@illinois.edu

## Abstract

Accurately answering a question about a given image requires combining observations with general knowledge. While this is effortless for humans, reasoning with general knowledge remains an algorithmic challenge. To advance research in this direction a novel 'fact-based' visual question answering (FVQA) task has been introduced recently along with a large set of curated facts which link two entities, *i.e.*, two possible answers, via a relation. Given a question-image pair, deep network techniques have been employed to successively reduce the large set of facts until one of the two entities of the final remaining fact is predicted as the answer. We observe that a successive process which considers one fact at a time to form a local decision is sub-optimal. Instead, we develop an entity graph and use a graph convolutional network to 'reason' about the correct answer by jointly considering all entities. We show on the challenging FVQA dataset that this leads to an improvement in accuracy of around 7% compared to the state of the art.

## 1 Introduction

When answering questions about images, we easily combine the visualized situation with general knowledge that is available to us. However, for algorithms, an effortless combination of general knowledge with observations remains challenging, despite significant work which aims to leverage these mechanisms for autonomous agents and virtual assistants.

In recent years, a significant amount of research has investigated algorithms for visual question answering (VQA) [2, 19, 32, 43, 44, 63], visual question generation (VQG) [21, 30, 36, 49], and visual dialog [12, 13, 20], paving the way to autonomy for artificial agents operating in the real world. Images and questions in these datasets cover a wide range of perceptual abilities such as counting, object recognition, object localization, and even logical reasoning. However, for many of these datasets the questions can be answered solely based on the visualized content, *i.e.*, no general knowledge is required. Therefore, numerous approaches address VQA, VQG and dialog tasks by extracting visual cues using deep network architectures [1, 2, 11, 16, 18, 19, 24, 32, 33, 35, 41, 44, 53, 56, 57, 59, 60, 63, 65, 66, 68], while general knowledge remains unavailable.

To bridge this discrepancy between human behavior and present day algorithmic design, Wang *et al*. [50] introduced a novel 'fact-based' VQA (FVQA) task, and an accompanying dataset containing images, questions with corresponding answers and a knowledge base (KB) of facts extracted from three different sources: WebChild [47], DBPedia [3] and ConceptNet [45]. Unlike classical VQA datasets, a question in the FVQA dataset is answered by a collective analysis of the information in the image and the KB of facts. Each question is mapped to a single supporting fact which contains the answer to the question. Thus, answering a question requires analyzing the image and choosing the right supporting fact, for which Wang *et al*. [50] propose a keyword-matching technique. This approach suffers when the question doesn't focus on the most obvious visual concept and when there are synonyms and homographs. Moreover, special information about the visual concept type and the answer source make it hard to generalize their approach to other datasets. We addressed these issues in our previous work [38], where we proposed a learning-based approach which embeds the image question pairs and the facts to the same space and ranks the facts according to their relevance. We observed a significant improvement in performance which motivated us to explore other learning based methods, particularly those which exploit the graphical structure of the facts.

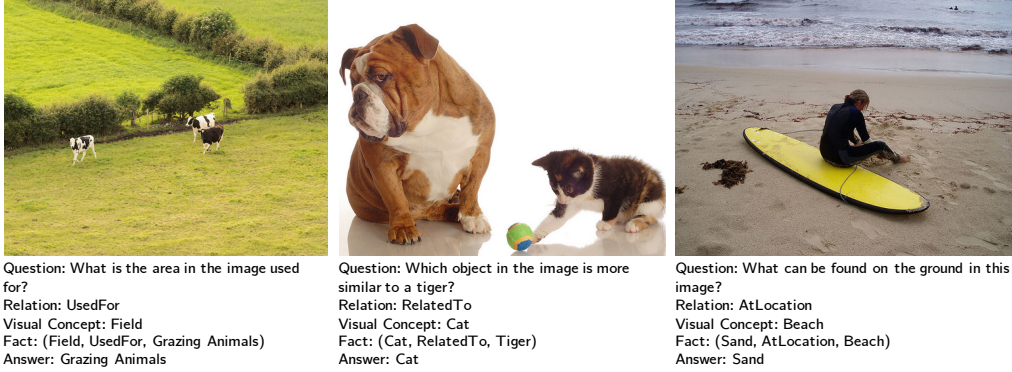

Question: What is the area in the image used for?
Relation: UsedFor
Visual Concept: Field
Fact: (Field, UsedFor, Grazing Animals)
Answer: Grazing Animals

Question: Which object in the image is more similar to a tiger?
Relation: RelatedTo
Visual Concept: Cat
Fact: (Cat, RelatedTo, Tiger)
Answer: Cat

Question: What can be found on the ground in this image?
Relation: AtLocation
Visual Concept: Beach
Fact: (Sand, AtLocation, Beach)
Answer: Sand

Figure 1: Results of our graph convolutional net based approach on the recently introduced FVQA dataset.

In this work, our main motivation is to develop a technique which uses the information from multiple facts before arriving at an answer and relies less on retrieving the single 'correct' fact needed to answer a question. To this end, we develop a model which 'thinks out of the box,' *i.e.*, it 'reasons' about the right answer by taking into account a list of facts via a Graph Convolution Network (GCN) [25]. The GCN enables *joint* selection of the answer from a list of candidate answers, which sets our approach apart from the previous methods that assess one fact at a time. Moreover, we select a list of supporting facts in the KB by ranking GloVe embeddings. This handles challenges due to synonyms and homographs and also works well with questions that don't focus on the main object.

We demonstrate the proposed algorithm on the FVQA dataset [50], outperforming the state of the art by around 7%. Fig. 1 shows results obtained by our model. Unlike the models proposed in [50], our method does not require any information about the ground truth fact (visual concept type and answer source). In contrast to our approach in [38], which focuses on learning a joint image-question-fact embedding for retrieving the right fact, our current work uses a simpler method for retrieving multiple candidate facts (while still ensuring that the recall of the ground truth fact is high), followed by a novel GCN inference step that collectively assesses all the relevant facts before arriving at an answer. Using an ablation analysis we find improvements due to the GCN component, which exploits the graphical structure of the knowledge base and allows for sharing of information between possible answers, thus improving the explainability of our model.

## 2 Related Work

We develop a visual question answering algorithm based on graph convolutional nets which benefits from general knowledge encoded in the form of a knowledge base. We therefore briefly review existing work in the areas of visual question answering, fact-based visual question answering and graph convolutional networks.

**Visual Question Answering:** Recently, there has been significant progress in creating large VQA datasets [2, 17, 23, 34, 41, 66] and deep network models which correctly answer a question about an image. The initial VQA models [1, 2, 4, 11, 16, 17, 19, 24, 32, 33, 35, 41, 44, 53, 57, 59, 60, 65, 68] combined the LSTM encoding of the question and the CNN encoding of the image using a deep network which finally predicted the answer. Results can be improved with attention-based multi-modal networks [1, 11, 16, 32, 43, 44, 57, 59] and dynamic memory networks [22, 56]. All of these methods were tested on standard VQA datasets where the questions can solely be answered by observing the image. No out of the box thinking was required. For example, given an image of a cat, and the question, "Can the animal in the image be domesticated?," we want our method to combine features from the image with common sense knowledge (a cat can be domesticated). This calls for the development of a model which leverages external knowledge.

**Fact-based Visual Question Answering:** Recent research in using external knowledge for natural language comprehension led to the development of semantic parsing [5, 6, 10, 15, 28, 31, 38, 40, 46, 55, 61, 62, 64] and information retrieval [7–9, 14, 26, 48, 58] methods. However, knowledge based visual question answering is fairly new. Notable examples in this direction are works by Zhu *et al*. [67], Wu *et al*. [54], Wang *et al*. [51], Narasimhan *et al*. [37], Krishnamurthy and Kollar [27], and our previous work, Narasimhan and Schwing [38].

*Ask Me Anything* (AMA) by Wu *et al*. [54], AHAB by Wang *et al*. [51], and FVQA by Wang *et al*. [50] are closely related to our work. In AMA, attribute information extracted from the image is used to query the external knowledge base DBpedia [3], to retrieve paragraphs which are summarized

to form a knowledge vector. The knowledge vector is combined with the attribute vector and multiple captions generated for the image, before being passed as input to an LSTM which predicts the answer. The main drawback of AMA is that it does not perform any explicit reasoning and ignores the possible structure in the KB. To address this, AHAB and FVQA attempt to perform explicit reasoning. In AHAB, the question is converted to a database query via a multistep process, and the response to the query is processed to obtain the final answer. FVQA also learns a mapping from questions to database queries through classifying questions into categories and extracting parts from the question deemed to be important. A matching score is computed between the facts retrieved from the database and the question, to determine the most relevant fact which forms the basis of the answer for the question. Both these methods use databases with a particular structure: facts are represented as tuples, for example, (*Apple*, *IsA*, *Fruit*), and (*Cheetah*, *FasterThan*, *Lion*).

The present work follows up on our earlier method, *Straight to the Facts* (STTF) [38]. STTF uses object, scene, and action predictors to represent an image and an LSTM to represent a question and combines the two using a deep network. The facts are scored based on the cosine similarity of the image-question embedding and fact embedding. The answer is extracted from the highest scoring fact.

We evaluate our method on the dataset released as part of the FVQA work, referred to as the FVQA dataset [50], which is a subset of three structured databases – DBpedia [3], ConceptNet [45], and WebChild [47].

**Graph Convolutional Nets:** Kipf and Welling [25] introduced Graph Convolutional Networks (GCN) to extend Conv nets (CNNs) [29] to arbitrarily connected undirected graphs. GCNs learn representations for every node in the graph that encodes both the local structure of the graph surrounding the node of interest, as well as the features of the node itself. At a graph convolutional layer, features are aggregated from neighboring nodes and the node itself to produce new output features. By stacking multiple layers, we are able to gather information from nodes further away. GCNs have been applied successfully for graph node classification [25], graph link prediction [42], and zero-shot prediction [52]. Knowledge graphs naturally lend themselves to applications of GCNs owing to the underlying structured interactions between nodes connected by relationships of various types. In this work, given an image and a question about the image, we first identify useful sub-graphs of a large knowledge graph such as DBpedia [3] and then use GCNs to produce representations encoding node and neighborhood features that can be used for answering the question.

Specifically, we propose a model that retrieves the most relevant facts to a question-answer pair based on GloVe features. The sub-graph of facts is passed through a graph convolution network which predicts an answer from these facts. Our approach has the following advantages: 1) Unlike FVQA and AHAB, we avoid the step of query construction and do not use the ground truth visual concept or answer type information which makes it possible to incorporate any fact space into our model. 2) We use GloVe embeddings for retrieving and representing facts which works well with synonyms and homographs. 3) In contrast to STTF, which uses a deep network to arrive at the right fact, we use a GCN which operates on a subgraph of relevant facts while retaining the graphical structure of the knowledge base which allows for reasoning using message passing. 4) Unlike previous works, we have reduced the reliance on the knowledge of the ground truth fact at training time.

## 3  Visual Question Answering with Knowledge Bases

To jointly 'reason' about a set of answers for a given question-image pair, we develop a graph convolution net (GCN) based approach for visual question answering with knowledge bases. In the following we first provide an overview of the proposed approach before delving into details of the individual components.

**Overview:** Our proposed approach is outlined in Fig. 2. Given an image $I$ and a corresponding question $Q$, the task is to predict an answer $A$ while using an external knowledge base KB which consists of facts, $f_i$, *i.e.*, KB $= \{f_1, f_2, \ldots, f_{|\text{KB}|}\}$. A fact is represented as a Resource Distribution Framework (RDF) triplet of the form $f = (x, r, y)$, where $x$ is a visual concept grounded in the image, $y$ is an attribute or phrase, and $r \in \mathcal{R}$ is a relation between the two entities, $x$ and $y$. The relations in the knowledge base are part of a set of 13 possible relations $\mathcal{R} = \{$*Category*, *Comparative*, *HasA*, *IsA*, *HasProperty*, *CapableOf*, *Desires*, *RelatedTo*, *AtLocation*, *PartOf*, *ReceivesAction*, *UsedFor*, *CreatedBy*$\}$. Subsequently we use $x(f)$, $y(f)$, or rel($f$) to extract the visual concept $x$, the attribute phrase $y$, or the relation $r$ in fact $f = (x, r, y)$ respectively.

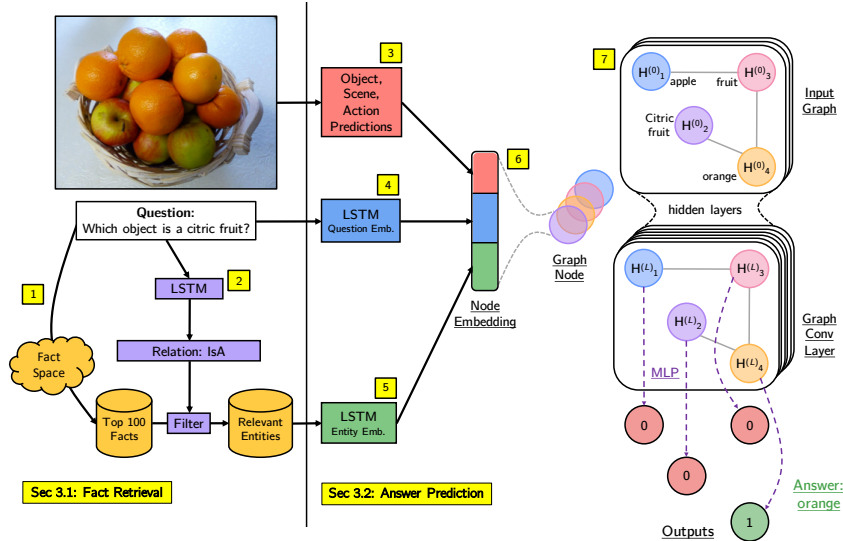

Figure 2: Outline of the proposed approach: Given an image and a question, we use a similarity scoring technique (1) to obtain relevant facts from the fact space. An LSTM (2) predicts the relation from the question to further reduce the set of relevant facts and its entities. An entity embedding is obtained by concatenating the visual concepts embedding of the image (3), the LSTM embedding of the question (4), and the LSTM embedding of the entity (5). Each entity forms a single node in the graph and the relations constitute the edges (6). A GCN followed by an MLP performs joint assessment (7) to predict the answer. Our approach is trained end-to-end.

Every question $Q$ is associated with a single fact, $f^*$, that helps answer the question. More specifically, the answer $A$ is one of the two entities of that fact, *i.e.*, either $A = x^*$ or $A = y^*$, both of which can be extracted from $f^* = (x^*, r^*, y^*)$.

Wang *et al.* [50] formulate the task as prediction of a fact $\hat{f} = (\hat{x}, \hat{r}, \hat{y})$ for a given question-image pair, and subsequently extract either $\hat{x}$ or $\hat{y}$, depending on the result of an answer source classifier. As there are over $190,000$ facts, retrieving the correct supporting fact $f^*$ is challenging and computationally inefficient. Usage of question properties like 'visual concept type' makes the proposed approach hard to extend.

Guided by the observation that the correct supporting fact $f^*$ is within the top-100 of a retrieval model $84.8\%$ of the time, we develop a two step solution: (1) retrieving the most relevant facts for a given question-image pair. To do this, we extract the top-100 facts, *i.e.*, $f_{100}$ based on word similarity between the question and the fact. Further, we obtain the set of relevant facts $f_{rel}$ by reducing $f_{100}$ based on consistency of the fact relation $r$ with a predicted relation $\hat{r}$. (2) predicting the answer as one of the entities in this reduced fact space $f_{rel}$. To predict the answer we use a GCN to compute representations of nodes in a graph, where the nodes correspond to the unique entities $e \in E = \{x(f) : f \in f_{rel}\} \cup \{y(f) : f \in f_{rel}\}$, *i.e.*, either $x$ or $y$ in the fact space $f_{rel}$. Two entities in the graph are connected if a fact relates the two. Using a GCN permits to jointly assess the suitability of all entities which makes our proposed approach different from classification based techniques.

For example, consider the image and the question shown in Fig. 2. The relation for this question is "IsA" and the fact associated with this question-image pair is (Orange, IsA, Citric). The answer is Orange. In the following we first discuss retrieval of the most relevant facts for a given question-image pair before detailing our GCN approach for extracting the answer from this reduced fact space.

## 3.1 Retrieval of Relevant Facts

To retrieve a set of relevant facts $f_{rel}$ for a given question-image pair, we pursue a score based approach. We first compute the cosine similarity of the GloVe embeddings of the words in the fact with the words in the question and the words of the visual concepts detected in the image. Because some words may differ between question and fact, we obtain a fact score by averaging the Top-K word similarity scores. We rank the facts based on their similarity and retrieve the top-100 facts for each question, which we denote $f_{100}$. We chose 100 facts as this gives the best downstream accuracy as shown in Tab. 1. As indicated in Tab. 1, we observe a high recall of the ground truth fact in the retrieved facts while using this technique. This motivates us to avoid a complex model which finds the right fact, as used in [50] and [38], and instead use the retrieved facts to directly predict the answer.

| | @1 | @50 | @100 | @150 | @200 | @500 |
|---|---|---|---|---|---|---|
| **Fact Recall** | 22.6 | 76.5 | 84.8 | 88.4 | 91.6 | 93.1 |
| **Downstream Accuracy** | 22.6 | 58.93 | **69.35** | 68.23 | 65.61 | 60.22 |

Table 1: Recall and downstream accuracy for different number of facts.

We further reduce this set of 100 facts by assessing their relation attribute. To predict the relation from a given question, we use the approach described in [38]. We retain the facts among the top-100 only if their relation agrees with the predicted relation $\hat{r}$, *i.e.*, $f_{\text{rel}} = \{f \in f_{100} : \text{rel}(f) = \hat{r}\}$.

For every question, unique entities in the facts $f_{\text{rel}}$ are grouped into a set of candidate entities, $E = \{x(f) : f \in f_{\text{rel}}\} \cup \{y(f) : f \in f_{\text{rel}}\}$, with $|E| \leq 200$ (2 entities/fact and at most 100 facts).

Currently, we train the relation predictor's parameters independently of the remaining model. In future work we aim for an end-to-end model which includes this step.

### 3.2 Answer Prediction

Given the set of candidate entities $E$, we want to 'reason' about the answer, *i.e.*, we want to predict an entity $\hat{e} \in E$. To jointly assess the suitability of all candidate entities in $E$, we develop a Graph-Convolution Net (GCN) based approach which is augmented by a multi-layer perceptron (MLP). The nodes in the employed graph correspond to the available entities $e \in E$ and their node representation is given as an input to the GCN. The GCN combines entity representations in multiple iterative steps. The final transformed entity representations learned by the GCN are then used as input in an MLP which predicts a binary label, *i.e.*, $\{1, 0\}$, for each entity $e \in E$, indicating if $e$ is or isn't the answer.

More formally, the goal of the GCN is to learn how to combine representations for the nodes $e \in E$ of a graph, $\mathcal{G} = (E, \mathcal{E})$. Its output feature representations depend on: (1) learnable weights; (2) an adjacency matrix $A_{\text{adj}}$ describing the graph structure $\mathcal{E}$. We consider two entities to be connected if they belong to the same fact; (3) a parametric input representation $g_w(e)$ for every node $e \in E$ of the graph. We subsume the original feature representations of all nodes in an $|E| \times D$-dimensional feature matrix $H^{(0)} \in \mathbb{R}^{|E| \times D}$, where $D$ is the number of features. In our case, each node $e \in E$ is represented by the concatenation of the corresponding image, question and entity representation, *i.e.*, $g_w(e) = (g_w^V(I), g_w^Q(Q), g_w^C(e))$. Combining the three representations ensures that each node/entity depends on the image and the question. The node representation is discussed in detail below.

The GCN consists of $L$ hidden layers where each layer is a non-linear function $f(\cdot, \cdot)$. Specifically,

$$H^{(l)} = f(H^{(l-1)}, A) = \sigma(\tilde{D}^{-1/2} \tilde{A} \tilde{D}^{-1/2} H^{(l-1)} W^{(l-1)}) \quad \forall l \in \{1, \ldots, L\}, \quad (1)$$

where the input to the GCN is $H^{(0)}$, $\tilde{A} = A_{\text{adj}} + I$ ($I$ is an identity matrix), $\tilde{D}$ is the diagonal node degree matrix of $\tilde{A}$, $W^{(l)}$ is the matrix of trainable weights at the $l$-th layer of the GCN, and $\sigma(\cdot)$ is a non-linear activation function. We let the $K$-dimensional vector $\hat{g}(e) \in \mathbb{R}^K$ refer to the output of the GCN, extracted from $H^{(L)} \in \mathbb{R}^{|E| \times K}$. Hereby, $K$ is the number of output features.

The output of the GCN, $\hat{g}(e)$ is passed through an MLP to obtain the probability $p_w^{\text{NN}}(\hat{g}(e))$ that $e \in E$ is the answer for the given question-image pair. We obtain our predicted answer $\hat{A}$ via

$$\hat{A} = \arg \max_{e \in E} p_w^{\text{NN}}(\hat{g}(e)). \quad (2)$$

As mentioned before, each node $e \in E$ is represented by the concatenation of the corresponding image, question and entity representation, *i.e.*, $g_w(e) = (g_w^V(I), g_w^Q(Q), g_w^C(e))$. We discuss those three representations subsequently.

**1. Image Representation:** The image representation, $g_w^V(I) \in \{0, 1\}^{1176}$ is a multi-hot vector of size 1176, indicating the visual concepts which are grounded in the image. Three types of visual concepts are detected in the image: actions, scenes and objects. These are detected using the same pre-trained networks described in [38].

**2. Question Representation:** An LSTM net is used to encode each question into the representation $g_w^Q(Q) \in \mathbb{R}^{128}$. The LSTM is initialized with GloVe embeddings [39] for each word in the question, which is fine-tuned during training. The hidden representation of the LSTM constitutes the question encoding.

**3. Entity Representation:** For each question, the entity encoding $g_w^C(e)$ is computed for every entity $e$ in the entity set $E$. Note that an entity $e$ is generally composed of multiple words. Therefore, similar

| Method | Accuracy | |
|---|---|---|
| | @1 | @3 |
| LSTM-Question+Image+Pre-VQA [50] | 24.98 | 40.40 |
| Hie-Question+Image+Pre-VQA [50] | 43.41 | 59.44 |
| FVQA [50] | 56.91 | 64.65 |
| Ensemble [50] | 58.76 | - |
| Straight to the Facts (STTF) [38] | 62.20 | 75.60 |

| Ours | Q | VC | Entity | MLP | GCN Layers | Rel | @1 | @3 |
|---|---|---|---|---|---|---|---|---|
| 1 | ✓ | - | ✓ | ✓ | - | - | 10.32 | 13.15 |
| 2 | ✓ | - | ✓ | ✓ | - | @1 | 13.89 | 16.40 |
| 3 | ✓ | - | ✓ | ✓ | 2 | @1 | 14.12 | 17.75 |
| 4 | ✓ | ✓ | ✓ | ✓ | - | - | 29.72 | 35.38 |
| 5 | ✓ | ✓ | ✓ | ✓ | - | @1 | 50.36 | 56.21 |
| 6 | ✓ | ✓ | ✓ | - | 2 | @1 | 48.43 | 53.87 |
| 7 | ✓ | ✓ | ✓ | ✓ | 1 | @1 | 54.60 | 60.91 |
| 8 | ✓ | ✓ | ✓ | ✓ | 1 | @3 | 57.89 | 65.14 |
| 9 | ✓ | ✓ | ✓ | ✓ | 3 | @1 | 56.90 | 62.32 |
| 10 | ✓ | ✓ | ✓ | ✓ | 3 | @3 | 60.78 | 68.65 |
| 11 | ✓ | ✓ | ✓ | ✓ | 2 | @1 | 65.80 | 77.32 |
| 12 | ✓ | ✓ | ✓ | ✓ | 2 | @3 | **69.35** | **80.25** |
| 13 | ✓ | ✓ | ✓ | ✓ | 2 | gt | 72.97 | 83.01 |
| Human | | | | | | | 77.99 | - |

Table 2: Answer accuracy over the FVQA dataset.

to the question encoding, the hidden representation of an LSTM net is used. It is also initialized with the GloVe embeddings [39] of each word in the entity, which is fine-tuned during training.

The answer prediction model parameters consists of weights from the question embedding, entity embedding, GCN, and MLP. These are trained end-to-end.

### 3.3 Learning

We note that the answer prediction and relation prediction model parameters are trained separately. The dataset, $\mathcal{D} = \{(I, Q, f^*, A^*)\}$, to train both these parameters is obtained from [50]. It contains tuples $(I, Q, f^*, A^*)$ each composed of an image $I$, a question $Q$, as well as the ground-truth fact $f^*$ and answer $A^*$.

To train the relation predictor's parameters we use the subset $\mathcal{D}_1 = \{(Q, r^*)\}$, containing pairs of questions and the corresponding relations $r^* = \text{rel}(f^*)$ extracted from the ground-truth fact $f^*$. Stochastic gradient descent and classical cross-entropy loss are used to train the classifier.

The answer predictor's parameters, consist of the question and entity embeddings, the two hidden layers of the GCN, and the layers of the MLP. The model operates on question-image pairs and extracts the entity label from the ground-truth answer $A^*$ of the dataset $\mathcal{D}$, $i.e.$, 0 if it isn't the answer and 1 if it is. Again we use stochastic gradient descent and binary cross-entropy loss.

## 4 Experimental Evaluation

Before assessing the proposed approach subsequently, we first review properties of the FVQA dataset. We then present quantitative results to compare our proposed approach with existing baselines before illustrating qualitative results.

**Factual visual question answering dataset:** To evaluate our model, We use the publicly available FVQA [50] knowledge base and dataset. This dataset consists of 2,190 images, 5,286 questions, and 4,126 unique facts corresponding to the questions. The knowledge base consists of 193,449 facts, which were constructed by extracting top visual concepts for all images in the dataset and querying for those concepts in the knowledge bases, WebChild [47], ConceptNet [45], and DBPedia [3].

**Retrieval of Relevant Facts:** As described in Sec. 3.1, a similarity scoring technique is used to retrieve the top-100 facts $f_{100}$ for every question. GloVe 100d embeddings are used to represent each word in the fact and question. An initial stop-word removal is performed to remove stop words (such as "what," "where," "the") from the question. To assign a similarity score to each fact, we compute the word-wise cosine similarity of the GloVe embedding of every word in the fact with the words in the question and the detected visual concepts. We choose the top K% of the words in the fact with the highest similarity and average these values to assign a similarity score to the fact. Empirically we

| Sub-component | Error % @1 |
|---|---|
| Fact-retrieval | 15.20 |
| Relation prediction | 9.4 |
| Answer prediction(GCN) | 6.05 |
| Total error | 30.65 |

Table 3: Error contribution of the sub-components of the model to the total Top-1 error (30.65%).

found $K = 80$ to give the best result. The facts are sorted based on the similarity and the 100 highest scoring facts are filtered. Tab. 1 shows that the ground truth fact is present in the top-100 retrieved facts 84.8% of the time and is retrieved as the top-1 fact 22.5% of the time. The numbers reported are an average over the five test sets. We also varied the number of facts retrieved in the first stage and report the recall and downstream accuracy in Tab. 1. The recall @50 (76.5%) is lower than the recall @100 (84.8%), which causes the final accuracy of the model to drop to 58.93%. When we retrieve 150 facts, recall is 88.4% and final accuracy is 68.23%, which is slightly below the final accuracy when retrieving 100 facts (69.35%). The final accuracy further drops as we increase the number of retrieved facts to 200 and 500.

**Predicting the relation:** As described earlier, we use the network proposed in [38] to determine the relation given a question. Using this approach, the Top-1 and Top-3 accuracy for relation prediction are 75.4% and 91.97% respectively.

**Predicting the Correct Answer:** Sec. 3.2 explains in detail the model used to predict an answer from the set of candidate entities $E$. Each node of the graph $\mathcal{G}$ is represented by the concatenation of the image, question, and entity embeddings. The image embedding $g_w^V(I)$ is a multi-hot vector of size 1176, indicating the presence of a visual concept in the image. The LSTM to compute the question embedding $g_w^Q(Q)$ is initialized with GloVe 100d embeddings for each of the words in the question. Batch normalization and a dropout of 0.5 is applied after both the embedding layer and the LSTM layer. The question embedding is given by the hidden layer of the LSTM and is of size 128. Each entity $e \in E$ is also represented by a 128 dimensional vector $g_w^C(e)$ which is computed by an LSTM operating on the words of the entity $e$. The concatenated vector $g_w(e) = (g_w^V(I), g_w^Q(Q), g_w^C(e))$ has a dimension of 1429 (*i.e.*, 1176+128+128).

For each question, the feature matrix $H^{(0)}$ is constructed from the node representations $g_w(e)$. The adjacency matrix $A_{\text{adj}}$ denotes the edges between the nodes. It is constructed by using the Top-1 or Top-3 relations predicted in Sec. 3.1. The adjacency matrix $A_{\text{adj}} \in \{0, 1\}^{200 \times 200}$ is of size $200 \times 200$ as the set $E$ has at most 200 unique entities (*i.e.*, 2 entities per fact and 100 facts per question). The GCN consists of 2 hidden layers, each operating on 200 nodes, and each node is represented by a feature vector of size 512. The representations of each node from the second hidden layer, *i.e.*, $H^{(2)}$ are used as input for a multi-layer perceptron which has 512 input nodes and 128 hidden nodes. The output of the hidden nodes is passed to a binary classifier that predicts 0 if the entity is not the answer and 1 if it is. The model is trained end-to-end over 100 epochs with batch gradient descent (Adam optimizer) using cross-entropy loss for each node. Batch normalization and a dropout of 0.5 was applied after each layer. The activation function used throughout is ReLU.

To prove the effectiveness of our model, we show six ablation studies in Tab. 2. Q, VC, Entity denote question, visual concept, and entity embeddings respectively. '11' is the model discussed in Sec. 3 where the entities are first filtered by the predicted relation and each node of the graph is represented by a concatenation of the question, visual concept, and entity embeddings. '12' uses the top three relations predicted by the question-relation LSTM net and retains all the entities which are connected by these three relations. '13' uses the ground truth relation for every question.

To validate the approach we construct some additional baselines. In '1,' each node is represented using only the question and the entity embeddings and the entities are not filtered by relation. Instead, all the entities in $E$ are fed to the MLP. '2' additionally filters based on relation. '3' introduces a 2-layer GCN before the MLP. '4' is the same as '1' except each node is now represented using question, entity and visual concept embeddings. '5' filters by relation and skips the GCN by feeding the entity representations directly to the MLP. '6' skips the MLP and the output nodes of the GCN are directly classified using a binary classifier. We observe that there is a significant improvement in performance when we include the visual concept features in addition to question and entity embeddings, thus highlighting the importance of the visual concepts. Without visual concepts, the facts retrieved in the first step have low recall which in turn reduces the downstream test accuracy.

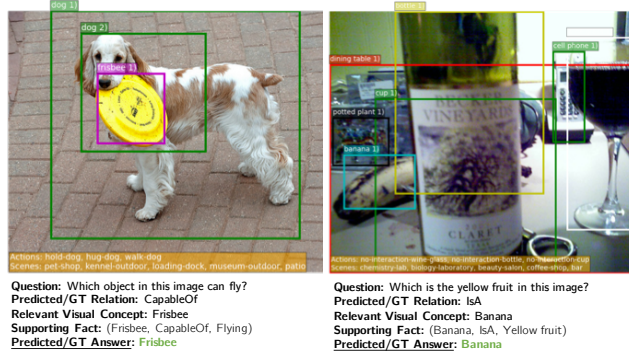

**Question:** Which object in this image can fly?
**Predicted/GT Relation:** CapableOf
**Relevant Visual Concept:** Frisbee
**Supporting Fact:** (Frisbee, CapableOf, Flying)
**Predicted/GT Answer: Frisbee**

**Question:** Which is the yellow fruit in this image?
**Predicted/GT Relation:** IsA
**Relevant Visual Concept:** Banana
**Supporting Fact:** (Banana, IsA, Yellow fruit)
**Predicted/GT Answer: Banana**

Figure 3: Visual Concepts (VCs) detected by our model. For each image we detect objects, scenes, and actions. We observe the supporting facts to have strong alignment with the VCs which proves the effectiveness of including VCs in our model.

We also report the top-1 and top-3 accuracy obtained by varying the number of layers in the GCN. With 3 layers ('9' and '10'), our model overfits, causing the test accuracy to drop to 60.78%. With 1 layer ('7' and '8'), the accuracy is 57.89% and we hypothesize that this is due to the limited information exchange that occurs with one GCN layer. We observe a correlation between the sparsity of the adjacency matrix and the performance of the 1 layer GCN model. When the number of facts retrieved is large and the matrix is less sparse, the 1 layer GCN model makes a wrong prediction. This indicates that the 2nd layer of the GCN allows for more message passing and provides a stronger signal when there are many facts to analyze.

We compare the accuracy of our model with the FVQA baselines and our previous work, STTF in Tab. 2. The accuracy reported here is averaged over all five train-test splits. As shown, our best model '13' outperforms the state-of-the-art STTF technique by more than 7% and the FVQA baseline without ensemble by over 12%. Note that combining GCN and MLP clearly outperforms usage of only one part. FVQA and STTF both try to predict the ground truth fact. If the fact is predicted incorrectly, the answer will also be wrong, thus causing the model to fail. Our method circumvents predicting the fact and instead uses multiple relevant facts to predict the answer. This approach clearly works better.

**Synonyms and homographs:** Here we show the improvements of our model compared to the baseline with respect to synonyms and homographs. To retrieve the top 100 facts, we use trainable word embeddings which are known to group synonyms and separate homographs.

We ran additional tests using Wordnet to determine the number of question-fact pairs which contain synonyms. The test data contains 1105 such pairs out of which our model predicts 95.38% correctly, whereas the FVQA and STTF models predict 78% and the 91.6% correctly. In addition, we manually generated 100 synonymous questions by replacing words in the questions with synonyms (*e.g.* "What in the bowl can you eat?", is rephrased as, "What in the bowl is edible?"). Tests on these 100 new samples find that our model predicts 91 of these correctly, whereas the key-word matching FVQA technique gets only 61 of these right. As STTF also uses GloVe embeddings, it gets 89 correct. With regards to homographs, the test set has 998 questions which contain words that have multiple meanings across facts. Our model predicts correct answers for 81.16%, whereas the FVQA model and STTF model get 66.33% and 79.4% correct, respectively.

**Qualitative results:** As described in Sec. 3.2, the image embedding is constructed based on the visual concepts detected in the image. Fig. 3 shows the object, scene, and action detection for two examples in our dataset. We also indicate the question corresponding to the image, the supporting fact, relation, and answer detected by our model. Using the high-level features helps summarize the salient content in the image as the facts are closely related to the visual concepts. We observe our model to work well even when the question does not focus on the main visual concept in the image. Tab. 2 shows that including the visual concept improves the accuracy of our model by nearly 20%.

Fig. 4 depicts a few success and failure examples of our method. In our model, predicting the correct answer involves three main steps: (1) Selecting the right supporting fact in the Top-100 facts, $f_{100}$; (2) Predicting the right relation; (3) Selecting the right entity in the GCN. In the top two rows of examples, our model correctly executes all the three steps. As shown, our model works for visual concepts of all three types, *i.e.*, actions, scenes and objects. Examples in the second row indicates that our model works well with synonyms and homographs as we use GloVe embeddings of words. The second example in the second row shows that our method obtains the right answer even when the

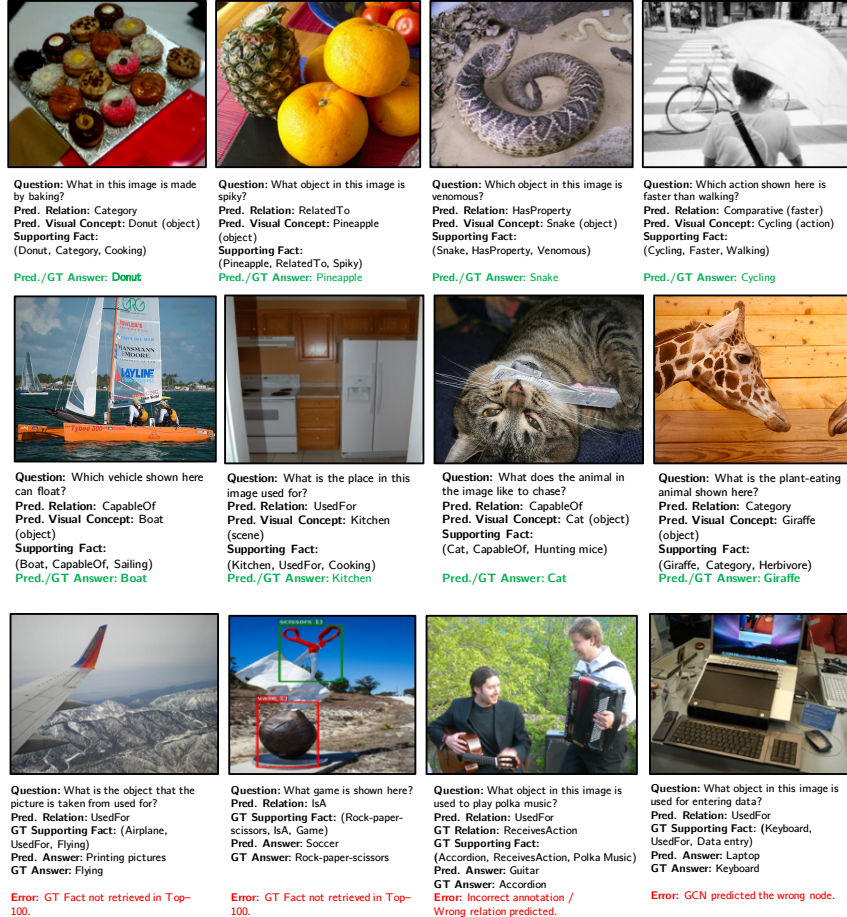

Figure 4: Success and failure cases: Success cases are shown in the top two rows. Our method correctly predicts the relation, visual concept, and the answer. The bottom row shows three different failure cases.

question and the fact do not have many words in common. This is due to the comparison with visual concepts while retrieving the facts.

The last row shows failure cases. Our method fails if any of the three steps produce incorrect output. In the first example the ground-truth fact (Airplane, UsedFor, Flying) isn't part of the top-100. This happens when words in the fact are neither related to the words in the question nor the list of visual concepts. A second failure mode is due to wrong node/entity predictions (selecting laptop instead of keyboard), *e.g.*, because a similar fact, (Laptop, UsedFor, Data processing) exists. These type of errors are rare (Tab. 3) and happen only when the fact space contains a fact similar to the ground truth one. The third failure mode is due to relation prediction accuracies which are around 75%, and 92% for Top-1 and Top-3 respectively, as shown in [38].

## 5 Conclusions

We developed a method for 'reasoning' in factual visual question answering using graph convolution nets. We showed that our proposed algorithm outperforms existing baselines by a large margin of 7%. We attribute these improvements to 'joint reasoning about answers,' which facilitates sharing of information before making an informed decision. Further, we achieve this high increase in performance by using only the ground truth relation and answer information, with no reliance on the ground truth fact. Currently, all the components of our model except for fact retrieval are trainable end-to-end. In the future, we plan to extend our network to incorporate this step into a unified framework.

**Acknowledgments:** This material is based upon work supported in part by the National Science Foundation under Grant No. 1718221 and Grant No. 1563727, Samsung, 3M, IBM-ILLINOIS Center for Cognitive Computing Systems Research (C3SR), Amazon Research Award, and AWS Machine Learning Research Award. We thank NVIDIA for providing the GPUs used for this research. We also thank Arun Mallya and Aditya Deshpande for their help.

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
