[Supplementary Material]

# Supplementary - Out of the Box: Reasoning with Graph Convolution Nets for Factual Visual Question Answering

**Medhini Narasimhan, Svetlana Lazebnik, Alexander G. Schwing**
University of Illinois Urbana-Champaign
{medhini2, slazebni, aschwing}@illinois.edu

Here we show a few more qualitative and quantitative results of our model. We also illustrate the improved explainability of our model.

## 1 Qualitative Results:

Fig. 1 demnostrates the GCN's ability to reason by sharing information between the nodes and also proves the explainability of our model. The circles indicate the nodes/entities. Green indicates our model voted for this node to be an answer(*i.e.* classified as 1). The darkness corresponds to the score it was assigned. We can see that the GT node has the darkest shade. Red indicates the node was classified 0.

Fig. 2 shows four cases where the method proposed by [1] fails but our model works. The figures show the visual concepts detected by our predictor as well.

Fig. 3 shows some more correct results obtained by our model.

Figure 1: Answer prediction using GCN.

**Question:** What can this place be used for?

**Relevant Scene:** Bathroom

**Predicted/GT Relation:** UsedFor

**Supporting Fact:** (Bathroom, UsedFor, Washing hands)

**Predicted/GT Answer:** Washing hands

**Question:** Which object in this image is utilized to chill food?

**Relevant Object:** Refrigerator

**Predicted/GT Relation:** UsedFor

**Supporting Fact:** (Refrigerator, UsedFor, Chilling food)

**Predicted/GT Answer:** Refrigerator

**Question:** What can I do using this place?

**Relevant Scene:** Kitchen

**Predicted/GT Relation:** UsedFor

**Supporting Fact:** (Kitchen, UsedFor, Cooking)

**Predicted/GT Answer:** Cooking

**Question:** What animal in this image can rest while standing?

**Relevant Object:** Horse

**Predicted/GT Relation:** CapableOf

**Supporting Fact:** (Horse, CapableOf, Rest standing up)

**Predicted/GT Answer:** Horse

Figure 2: Success cases.

## 2 Quantitative Results:

Table 1 shows the top-1 and top-3 accuracy in predicting the answers on questions which contain facts from different knowledge bases. All the facts in DBpedia belong to the class "Category". These are easy to identify as the question usually contains the term "Category". For example, the question, "What category of food does cake belong to?" has the supporting fact as "(Cake, Category, Herbivore)". Our model attains an Top-1 accuracy of 75.32% on this. ConceptNet consists of the 11 relations excluding "Category" and "Comparative." Identifying these relations from the question is pretty straightforward and we achieve a Top-1 accuracy of 73.71% in predicting the answer on questions that have facts from ConceptNet. The error in both ConceptNet and DBpedia is mainly due to some visual concepts going undetected. "Webchild" consists of facts with comparative terms (such as "faster", "stronger") which are also easy to identify in the given question. Our model accurately predicts the answer for these questions 56.22% of the time.

We also report the Wu-Palmer Similarity (WUPS) scores [2] in Tables 2 and 3. WUPS computes the similarity between two words based on their common subsequence in the taxonomy tree. If the similarity between the predicted and GT answer is greater than a certain threshold, the answer

**Question:** What is the name of this animal's clone?

**Relevant Object:** Sheep

**Predicted/GT Relation:** RelatedTo

**Supporting Fact:** (Sheep, RelatedTo, Clone Dolly)

**Predicted/GT Answer:** Dolly

**Question:** What category of food does coke belong to?

**Relevant Question Keyword:** Coke

**Predicted/GT Relation:** Category

**Supporting Fact:** (Coke, Category, Junk food)

**Predicted/GT Answer:** Coke

**Question:** Which object in this image can protect you from the sun?

**Relevant Object:** Umbrella

**Predicted/GT Relation:** CapableOf

**Supporting Fact:** (Umbrella, CapableOf, Shade from sun)

**Predicted/GT Answer:** Umbrella

**Question:** What object in the image can be used to lift food?

**Relevant Object:** Fork

**Predicted/GT Relation:** UsedFor

**Supporting Fact:** (Fork, UsedFor, Picking food up)

**Predicted/GT Answer:** Fork

Figure 3: Success cases.

predicted is considerred right. We report the WUPS at thresholds 0.9 and 0.0. Our model performs better than the state-of-the-art models at both thresholds.

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

| Method | KB-Source Accuracy | | | | | |
| --- | --- | --- | --- | --- | --- | --- |
| | DBpedia | | ConceptNet | | WebChild | |
| | Top-1 | Top-3 | Top-1 | Top-3 | Top-1 | Top-3 |
| LSTM-Question+Image+Pre-VQA [1] | 15.38 | 32.64 | 25.97 | 41.02 | 34.42 | 50.27 |
| Hie-Question+Image+Pre-VQA [1] | 38.39 | 56.07 | 43.23 | 59.47 | 52.85 | 66.49 |
| FVQA, top-1 [1] | 51.25 | 63.07 | 53.50 | 60.16 | 43.54 | 46.58 |
| FVQA, top-3 [1] | 56.67 | 69.31 | 57.60 | 64.70 | 48.74 | 53.45 |
| Ensemble [1] | 57.08 | - | 58.98 | - | 59.77 | - |
| | | | | | | |
| Ours - Q_VC_Entity + GCN + MLP + Rel@1 | 72.97 | 80.76 | 71.20 | 75.82 | 53.33 | 68.69 |
| **Ours - Q_VC_Entity + GCN + MLP + Rel@3** | **75.32** | **83.91** | **73.71** | **79.64** | **56.22** | **71.39** |
| Ours - Q_VC_Entity + GCN + MLP + *gt*-Rel | 83.24 | 90.23 | 80.76 | 86.90 | 64.78 | 81.31 |
| Human | 74.41 | - | 78.32 | - | 81.95 | - |

Table 1: Accuracy in predicting the correct answer based on the knowledge base(KB). Best model is shown in bold.

| Method | WUPS@0.9 | |
| --- | --- | --- |
| | @1 | @3 |
| LSTM-Question+Image+Pre-VQA [1] | 31.96 | 48.55 |
| Hie-Question+Image+Pre-VQA [1] | 48.93 | 64.75 |
| FVQA, top-1 [1] | 54.79 | 61.41 |
| FVQA, top-3 [1] | 59.67 | 66.89 |
| Ours - Q_VC_Entity + GCN + MLP + Rel@1 | 64.70 | 65.12 |
| **Ours - Q_VC_Entity + GCN + MLP + Rel@3** | **69.63** | **71.28** |
| Ours - Q_VC_Entity + GCN + MLP + *gt*-Rel | 72.24 | 76.19 |
| Human | 82.47 | - |

Table 2: WUPS@0.9 over the FVQA dataset.

| Method | WUPS@0.0 | |
| --- | --- | --- |
| | @1 | @3 |
| LSTM-Question+Image+Pre-VQA [1] | 63.42 | 76.63 |
| FVQA, top-1 [1] | 64.96 | 69.57 |
| Hie-Question+Image+Pre-VQA [1] | 71.51 | 82.71 |
| FVQA, top-3 [1] | 72.34 | 77.52 |
| Ours - Q_VC_Entity + GCN + MLP + Rel@1 | 82.98 | 86 |
| **Ours - Q_VC_Entity + GCN + MLP + Rel@3** | **83.55** | **88.01** |
| Ours - Q_VC_Entity + GCN + MLP + *gt*-Rel | 85.97 | 90.34 |
| Human | 87.30 | - |

Table 3: WUPS@0.0 over the FVQA dataset.