[Reviews · NeurIPS 2018]

Reviewer 1



— This paper introduces a graph-convolutional-net-based approach for the task of factual visual question answering, where given an image and a question, the model has to retrieve the correct supporting knowledge fact to be able to answer it accurately. — Prior approaches query the knowledge base using a database query predicted from the question, and use the response from the database to obtain the final answer. This ignores the inherent graph structure of the knowledge base, and performs reasoning from facts to answer one at a time, which is computationally inefficient. — Instead, the proposed approach first has a coarse-to-fine candidate fact retrieval step which considers 1) similarity in GLoVE space between question and fact words, and then 2) filters this list to keep facts which align with the predicted relation from question. — The unique entities occurring in this filtered list of facts constitutes the nodes of a graph, with corresponding node representations as a concatenation of 1) image feature vector, 2) LSTM embedding of question, and 3) LSTM embedding of entity. Two entities have a connecting edge if they belong to the same fact. — This graph is operated on by a Graph Convolution Network with 2 hidden layers, and its output representation for each node is used as input to a binary classifier to predict if that entity is the answer or not. Strengths — The proposed approach is intuitive, sufficiently novel, and outperforms prior work by a large margin — ~10% better than the previous best approach, which is an impressive result. — Error breakdown by subcomponent is neat. — Overall, the paper is clearly written, and has fairly exhaustively done experiments and ablation studies. Weaknesses — Given that the fact retrieval step is still the bottleneck in terms of accuracy (Table 4), it would be useful to check how sensitive downstream accuracy is to the choice of retrieving 100 facts. What is the answering accuracy if 50 facts are retrieved? 500? Evaluation Overall, I’m happy to recommend this paper for publication. Using graph convolutional networks is an intuitive approach to reasoning over knowledge bases, and hasn’t been tried before for the task of FVQA. Going forward, I encourage the authors to push more on learning the entire network (relation prediction, answer prediction) jointly, which might lead to improved accuracies due to tighter coupling and end-to-end training.

Reviewer 2



[Summary] This paper proposed to use the graph convolution neural networks for factual visual question answering. The proposed method first retrieve the fact based on glove vector and filtered by relation predicted based on the question. The filtered fact concatenate with question embedding and other visual features are serve as graph nodes and further go through a 2 layer graph convolution network. The refined features are further fed in an MLP to predict the binary label for each entity. The authors performed experiments on FVQA dataset and achieve state-of-the-art performance compare with previous results. [Strength] 1: Experiment results on FVQA dataset is very good. 2: The proposed framework is interesting and seems general for the factual visual question answering task. [Weakness] 1: Poor writing and annotations are a little hard to follow. 2: Although applying GCN on FVQA is interesting, the technical novelty of this paper is limited. 3: The motivation is to solve when the question doesn't focus on the most obvious visual concept when there are synonyms and homographs. However, from the experiment, it's hard to see whether this specific problem is solved or not. Although the number is better than the previous method, it will be great if the authors could product more experiments to show more about the question/motivation raised in the introduction. 4: Following 3, applying MLP after GCN is very common, and I'm not surprised that the performance will drop without MLP. The authors should show more ablation studies on performance when varying the number of facts retrieval, what happened if we different number of layer of GCN?

Reviewer 3



This paper proposes a neural architecture for Fact Based VQA (introduced in [1]), an extension of VQA requiring leveraging of facts from an additional knowledge base. Given an image and a question about it, the model further retrieves the top-100 facts from the KB based on cosine similarity between an averaged Glove embedding representation between question and fact. Further, the facts are embedded as entities in a graph, and each entity is processed via a Graph Convolutional Network and fed to a multilayer perceptron trained to be verified as the correct answer. Results, comparisons against baselines, and ablations are presented on the FVQA dataset. Strengths – This is a well written and interesting work that approaches an important problem and presents a simple and effective model for the same – The approach significantly outperforms the previous state of the art on Fact based VQA, and the ablations studied and error contribution analysis validate the modeling choices effectively. In general, the experimental results are quite thorough and promising, and the qualitative results are compelling and provide good insights. – The paper provides sufficient experimental detail and the results should be possible to reproduce Weaknesses/Questions – In the error decomposition, the fact-retrieval error can be controlled for by setting a higher threshold at the cost of more computation. Have the authors experimented with retrieving a larger set of facts, or were diminishing returns observed after the threshold of 100? – Some additional analysis of the typical structure of the induced graph would be interesting to see – how sparse is the adjacency matrix on average? L95 states: ‘By stacking multiple layers, we are able to gather information from nodes further away’ – some analysis/empirical evidence of whether the model can pick up on longer range relationships in the graph, would also be interesting to include and would strengthen the paper. [1] Wang, Peng, Qi Wu, Chunhua Shen, Anthony Dick, and Anton van Den Hengel. 2017. “FVQA: Fact-Based Visual Question Answering.” IEEE TPAMI ---- I have read through the author response and reviews, and am happy to keep my accept rating. The paper is well written and presents a novel formulation and strong results for the FVQA task. The analysis and experimentation is thorough, and in my opinion the authors have provided a strong rebuttal to most concerns raised in the reviews.